# Prognostic factors of chronic pulmonary aspergillosis: A retrospective cohort of 264 patients from Japan

Yuya Kimura*, Yuka Sasaki, Junko Suzuki, Jun Suzuki, Hiroshi Igei, Maho Suzukawa, Hirotoshi Matsui

Center for Pulmonary Diseases, Department of Respiratory Medicine, National Hospital Organization Tokyo National Hospital, Tokyo, Japan

* yuk.close.to.wrd.34@gmail.com

## Abstract

### Background

Chronic pulmonary aspergillosis (CPA) develops in various underlying pulmonary conditions. There is scarce data evaluating interstitial lung disease (ILD)/abnormalities (ILA) as such conditions, and it has not been explored much whether non-tuberculous mycobacterial pulmonary disease (NTM-PD) is a prognostic factor for mortality in CPA patients. Few reports had investigated prognostic factors of CPA including underlying pulmonary conditions.

### Objectives

To explore prognostic factors of CPA including pulmonary conditions.

### Methods

We conducted a retrospective cohort study of 264 CPA patients from a center for pulmonary aspergillosis in Japan.

### Results

Survival rates were 78.7%, 61.0%, and 47.4% at 1, 3, and 5 years, respectively. Of 264 patients, 53 (20.1%) and 87 (33.1%) were complicated with ILA and NTM-PD. Several independent prognostic factors were identified by multivariate Cox proportional analysis: ILA (HR 1.76, 95%CI 1.06–2.92, p = 0.029), age (1.05, 1.02–1.08, p<0.001), male sex (2.48, 1.34–4.59, p = 0.004), body mass index of <18.5 kg/m$^2$ (1,87, 1.20–2.90, p = 0.005), presence of aspergilloma (1.59, 1.04–2.45, p = 0.033), and lower serum albumin (0.56, 0.38–0.83, p = 0.004). NTM-PD was not associated with higher mortality (0.85, 0.52–1.38, p = 0.51).

**Data Availability Statement:** All relevant data are within the paper and its Supporting Information files.

**Funding:** The authors received no specific funding for this work.

**Competing interests:** The authors have declared that no competing interests exist.

## Conclusions

The poor prognosis of CPA and several prognostic factors were revealed. Early diagnosis and intervention is required with reference to such factors.

## Introduction

Chronic pulmonary aspergillosis (CPA) is a chronic progressive infectious disease caused by the *Aspergillus* species that currently affects an estimated three million people globally [1]. The prognosis is poor, with a 5-year survival rate ranging from 17.5% to 85% [2–8], and it is only recently that this infectious disease has been recognized as a significant global health burden [9].

Patients with CPA have a variety of underlying pulmonary conditions, such as previous tuberculosis, non-tuberculous mycobacterial pulmonary disease (NTM-PD), chronic obstructive pulmonary disease and/or emphysema [10]. It is easily imagined that the kind of underlying pulmonary condition can affect the prognosis of CPA patients. Common underlying pulmonary conditions include chronic obstructive pulmonary disease and previous tuberculosis, but patients with interstitial lung abnormalities (ILA)/disease (ILD) also have a potential risk for CPA. However, there is a paucity of data evaluating the complication of ILA/ILD, although previous prognostic studies of CPA that included ILD in the underlying pulmonary conditions have reported 2–17% patients were associated with ILD [3, 5, 6, 11, 12]. In addition, the effect of underlying NTM-PD on the prognosis of CPA patients has not been fully explored, even though previous studies showed that complication of CPA was an independent risk factor for mortality in non-tuberculous mycobacterium (NTM) patients [13–16].

Up to now, some independent prognostic factors were revealed, such as lower body mass index (BMI) [6, 11] and lower albumin [8, 11], but scarce reports had included underlying pulmonary conditions in candidates of the prognostic factors. Interestingly, the distribution of underlying conditions of CPA is different dependent on the region of the world, so data of the prognostic factors from different areas is desired.

We present the long-term prognosis of 264 newly-diagnosed CPA patients from an experienced center for pulmonary aspergillosis in Japan. In addition, the prognostic factors including underlying pulmonary conditions are shown.

## Materials and methods

### Study patients

We reviewed the medical records of 1,141 patients with positive serum anti-*Aspergillus* antibody obtained between January 2012 and April 2017 at the National Hospital Organization Tokyo Hospital, Tokyo, Japan. Of these, 359 patients were newly diagnosed with CPA. The diagnosis was confirmed by at least two respiratory physicians with at least 15 years of experience using the criteria modified by DENNING et al. [17] and described in detail by FARID et al. [18] and the European guideline on CPA [19]. We had limited CPA cases to cases with a confirmed positive serum anti-*Aspergillus* antibody, so the criteria were as follows: 1) clinical symptoms or progressive radiological abnormality lasting for at least 3 months (containing inferred cases), 2) the presence of aspergilloma or one or more cavities consistent with CPA, and 3) serum anti-*Aspergillus* antibody positive. Serum anti-*Aspergillus* antibody was measured via Ouchterlony methods by the appearance of visible precipitations in a diffusion gel [20, 21]. The antibody was judged positive when the serum reacted with either 2mg/ml of *A*.

*fumigatus* culture filtrate antigen or 20 mg/ml of *A.fumigatus* somatic antigen. We excluded the following patients; 1) patients with simple pulmonary aspergilloma/*Aspergillus* nodule, 2) patients with a possible risk of subacute invasive pulmonary aspergillosis, *i.e.* who had received over 15 mg/day of prednisolone or a corresponding amount of systemic corticosteroid, with any hematological malignancy, and a positive HIV antibody test or whose symptoms were obviously acute. There were no patients concurrent with histoplasmosis or coccidioidomycosis. Of the 359 patients with newly diagnosed CPA, we excluded 25 patients with concurrent active pulmonary tuberculosis infection, 20 patients with concurrent active cancer (under any treatment), and 34 patients who had undergone subsequent surgery related with CPA because such conditions can affect prognosis. Finally, 264 patients were included. A figure showing patient flow is provided in S1 Fig.

## Study design

Using medical records of newly diagnosed 264 CPA patients as of April 2020, a retrospective review of clinical data, including survival status, was conducted. Time zero was defined as the first detection date of serum anti-*Aspergillus* antibody positive. We checked clinical information including patients sex, age at time zero, BMI, smoking history, baseline corticosteroid use, auto immune diseases, serum albumin and C-reactive protein, underlying pulmonary conditions (previous tuberculosis, NTM, emphysema, bronchiectasis, ILA, pneumothorax, previous thoracic surgery) and the presence of aspergilloma. The data of the nearest point of time zero within a week before or after the point were adopted for serum data. The data within three months of time zero was adopted for body mass index (BMI).

This study was approved by the institutional review board of the National Hospital Organization Tokyo Hospital. The requirement for informed consent was waived owing to its retrospective nature.

## Underlying pulmonary conditions

Previous pulmonary tuberculosis infection was accepted in any of the following cases: 1) detected in the referral document, 2) mediastinal lymph node calcification in CT scan, and 3) interferon gamma release assay positive. NTM was confirmed according to The American Thoracic Society and Infectious Disease Society of America's diagnostic criteria for nontuberculous mycobacterial pulmonary infections [22]: imaging studies consistent with the pulmonary disease, and repeated isolation of mycobacteria from sputum or isolated from at least one bronchial wash. Of note, we limited NTM-complicated cases to those who fulfilled the above criteria before the diagnosis of CPA. Bronchiectasis that led to a cavitary lesion was not counted as a secondary change. ILA was categorized into the following five categories: UIP/NSIP/PPFE pattern/combined pulmonary fibrosis and emphysema/unclassifiable. (UIP was limited to HRCT findings of basal predominant honeycombing, and combined pulmonary fibrosis and emphysema was limited to HRCT findings of low attenuation area with basal predominant pulmonary fibrosis.) Previous thoracic surgery was counted if it was detected in the referral document or an obvious surgical change on CT scan. The computed tomography scan closest to time zero was adopted (almost all had been taken within a month of time zero) and reviewed by two respiratory physicians with more than 15 years of experience (Dr. Yuka Sasaki, Dr. Jun Suzuki).

## Statistical analyses

Normally distributed data is presented as mean ± standard deviation, and non-normally distributed as median, interquartile range. Patients' characteristics were compared using Fisher's

exact test or Chi square test for nominal variables and Mann Whitney test for continuous variables. Patient survival was estimated using a Kaplan-Meier analysis, and log-rank tests were used to compare survival between groups. Patients who were lost to follow-up were censored at the date of last contact. A Cox proportional hazards model was used to calculate the hazard ratio (HR) of a variable and presented as HR (95%CI). The candidate variables for being entered into multivariate analysis were selected with reference to previous studies about CPA prognosis: BMI [6, 11], serum albumin [8, 11], serum C-reactive protein [5], the presence of aspergilloma [8], and use of systemic corticosteroids [5]. In these candidate variables, those that have p value of <0.05 in univariate Cox proportional hazards model, were entered into multivariate Cox proportional hazards model, along with all underlying pulmonary conditions. Apart from that, prognostic values of underlying pulmonary conditions were assessed by log-rank test and cox proportional hazards model including age. A p value of <0.05 was considered significant for all statistical tests. All analyses were completed using R statistical software (version 4.0.2).

## Results

### Patient background

All the patients in this study were Asian, and the median age at time zero was 71.0±11.1 years, and 195 (73.9%) patients were male (Table 1). Of the underlying pulmonary conditions, previous tuberculosis comprised 156 (59.1%), followed by emphysema 115 (43.6%), NTM 87 (33.0%), and ILA 53 (20.1%). The breakdown of ILA patterns was as follows: UIP 10 (18.9%)/ NSIP 0 (0.0%)/PPFE pattern 14 (26.4%)/combined pulmonary fibrosis and emphysema 17 (32.1%)/unclassifiable 12 (22.6%). The median [interquartile range, range] number of underlying pulmonary conditions was 2 [1, 2, 0–4]. The rates of previous tuberculosis, and emphysema were higher in males, while in contrast, the rates of NTM and bronchiectasis were higher in females. Smoking rate was much higher in male than in female (88.7% vs 15.9%).

**Table 1. Baseline characteristics of all study patients at diagnosis (n = 264).**

|  | Male (195) | Female (69) |
|---|---|---|
| Age, years | 71.1±10.6 | 70.6±12.5 |
| Body mass index, kg/m$^2$ | 18.1 [16.2–20.7] | 16.8 [14.9–18.5] |
| Underlying pulmonary conditions |  |  |
| Previous pulmonary tuberculosis | 131 (67.2%) | 25 (36.2%) |
| Non-tuberculous mycobacterium | 47 (24.1%) | 40 (58.0%) |
| Emphysema | 110 (56.4%) | 5 (7.2%) |
| Pneumothorax | 8 (4.1%) | 1 (1.4%) |
| Bronchiectasis | 13 (6.7%) | 33 (47.8%) |
| Thoracic surgery | 26 (13.3%) | 7 (10.1%) |
| Interstitial lung abnormalities | 43 (22.1%) | 10 (14.5%) |
| Presence of aspergilloma | 71 (36.4%) | 17 (24.6%) |
| Smoking history | 173 (88.7%) | 11 (15.9%) |
| Auto immune diseases | 8 (4.1%) | 5 (7.2%) |
| Use of systemic corticosteroids | 15 (7.7%) | 2 (2.9%) |
| Albumin, g/dL | 3.2[2.7–3.8] | 3.5[3.0–3.9] |
| C-reactive protein, mg/dL | 3.06 [0.65–9.63] | 1.22 [0.31–6.58] |

Data are presented as no. (%) or median [interquartile range].

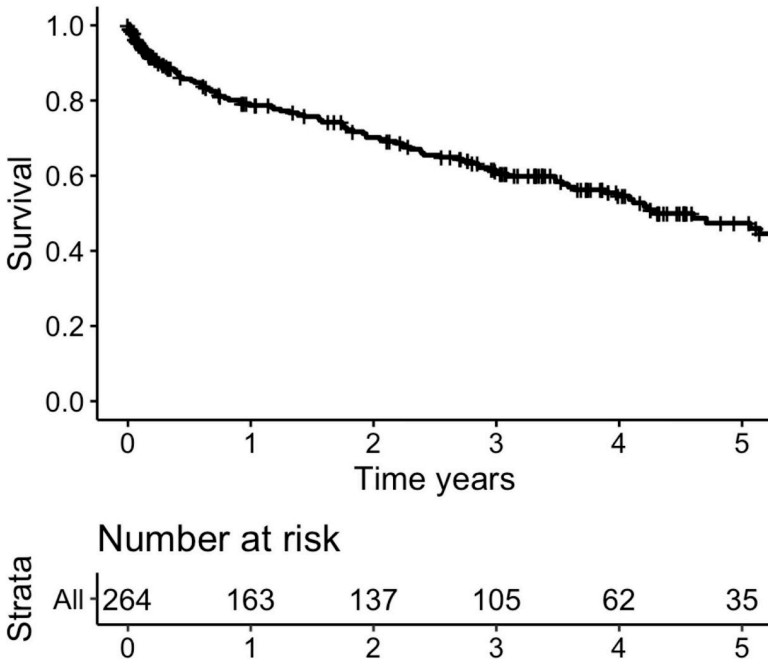

**Fig 1. Censored survival curve from the diagnosis of chronic pulmonary aspergillosis.** n = 264. 1-, 3-, 5-year survival 79%, 61%, 47%, respectively. The event had occurred in 103 patients.

## Overall survival

The overall observation period was a median of 803 days [115.5 days, 1435.5 days], with that of the survivors being 1,150 days [120 days, 1,634 days]. Of the total 264 patients, the event had occurred in 103 patients. Fig 1 shows the survival curve from diagnosis (time zero). The 1-, 3-, 5-year survival rates in our 264 patients were 78.7%, 61.0%, 47.4%, respectively.

## Prognostic factors

ILA was the only underlying condition associated with increased mortality in the log-rank test (p<0.001), and even in the Cox proportional hazards analysis including all the underlying conditions and age (HR 1.96, 95%CI 1.25–3.10, p = 0.004) (Table 2). The 1-, 3-, 5-year survival rates in 53 ILA-complicated patients were 67.2%, 37.6%, 13.8%, respectively (Fig 2). One or more aspergillomas were detected in 15 patients, out of whom nine were based on ILA findings (UIP 2, PPFE pattern 3, combined pulmonary fibrosis and emphysema 2, unclassifiable 2) (Fig 3).

In contrast, NTM was not related with increased mortality in the log-rank test (p = 0.81). The breakdown of etiologic NTM organisms was as follows: 38 (43.7%) *M. avium*, 20 (23.0%) *M. intracellulare*, 5 (5.7%) abscessus complex, and 24 (27.6%) other (8 *M. kansasii*, 3 *M. gordonae*, 1 *M. shinjukuense*, 1 *M. nonchromogenicum*, 1 *M. shimoidei*, 1 *M. xenopi*, and 9 unknown). The 2-year survival rate in patients with *M. avium* and *M. intracellulare* were 84.2 ±6.5%, and 60.4±11.8%, respectively (HR 1.78, 95%CI 0.74–4.25, p = 0.19). (See S2 Fig)

The results of the univariate Cox proportional hazards model of the surveyed variables are shown in Table 3. Seven significant risk factors for mortality except for underlying pulmonary conditions were identified: age, male sex, BMI of <18.5 kg/m$^2$, lower serum albumin, elevated C-reactive protein, presence of aspergilloma, and baseline corticosteroid use. Smoking history (HR 1.62, 95%CI 0.99–2.65, p = 0.054) and auto immune diseases (1.71, 0.79–3.71, p = 0.17) were not significantly correlated with mortality.

**Table 2. Underlying pulmonary conditions and their effect on survival of 264 CPA patient.**

| | Patients | Kaplan-Meier analysis | | Cox-proportional hazards model including all underlying conditions and age | |
|---|---|---|---|---|---|
| | | 2-year survival | | Log rank test p-value | Adjusted hazard ratio (95% CI) | p-value |
| **TB** | 156 [59.1] | with | 74.4±3.9 | 0.62 | 0.94 (0.61–1.46) | 0.79 |
| | | without | 64.7±5.0 | | | |
| **NTM** | 87 [33.0] | with | 72.5±5.3 | 0.81 | 0.95 (0.60–1.51) | 0.83 |
| | | without | 69.0±3.8 | | | |
| **Emphysema** | 115 [43.6] | with | 70.4±4.8 | 0.99 | 0.87 (0.57–1.32) | 0.51 |
| | | without | 69.9±4.1 | | | |
| **Pneumothorax** | 9 [3.4] | with | 55.6±16.6 | 0.28 | 1.36 (0.54–3.41) | 0.52 |
| | | without | 70.7±3.2 | | | |
| **Bronchiectasis** | 46 [17.4] | with | 79.3±6.6 | 0.22 | 0.73 (0.39–1.36) | 0.32 |
| | | without | 68.2±3.5 | | | |
| **Thoracic surgery** | 33 [12.5] | with | 72.0±8.5 | 0.51 | 1.20 (0.66–2.19) | 0.54 |
| | | without | 69.9±3.3 | | | |
| **ILA** | 53 [20.1] | with | 52.9±7.4 | **<0.001** | 1.96 (1.25–3.10) | **0.004** |
| | | without | 75.1±3.3 | | | |

Data are presented as n [%] or % ± SE. TB: tuberculosis; NTM: non-tuberculous mycobacteria; ILA: interstitial lung abnormalities. There were two CPA patients complicated with sarcoidosis, but it was too small to justify statistical analysis.

Multivariate analysis including all underlying pulmonary conditions, age, sex, BMI of <18.5 kg/m$^2$, presence of aspergilloma, serum albumin, C-reactive protein, and use of systemic corticosteroids showed that ILA (HR 1.76, 95%CI 1.06–2.92, p = 0.029), age (1.05, 1.02–1.08, p<0.001), male sex (2.48, 1.34–4.59, p = 0.004), BMI of <18.5 kg/m$^2$ (1,87, 1.20–2.90,

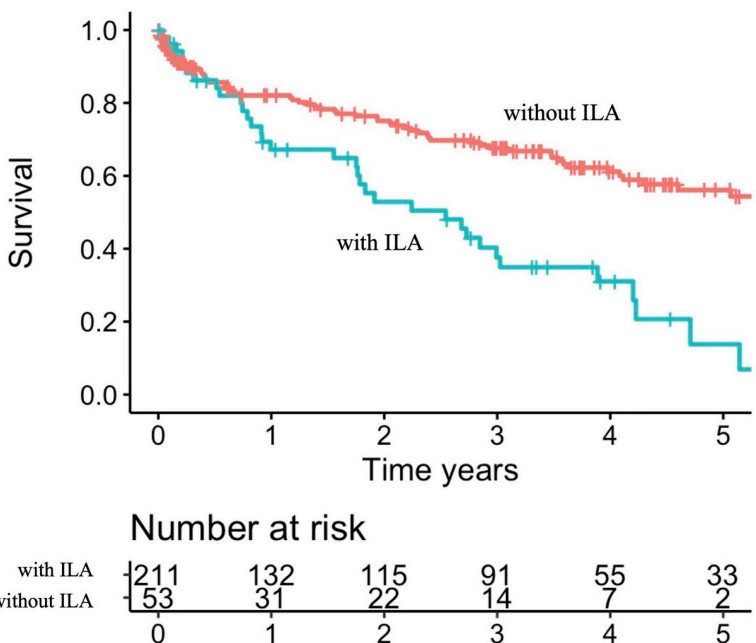

**Fig 2. Survival stratified by presence of Interstitial Lung Abnormalities (ILA).** 1-, 3-, 5-year survival rate without ILA 82%, 68%, 56%, respectively; 67%, 38%, 14%, respectively for those with ILA.

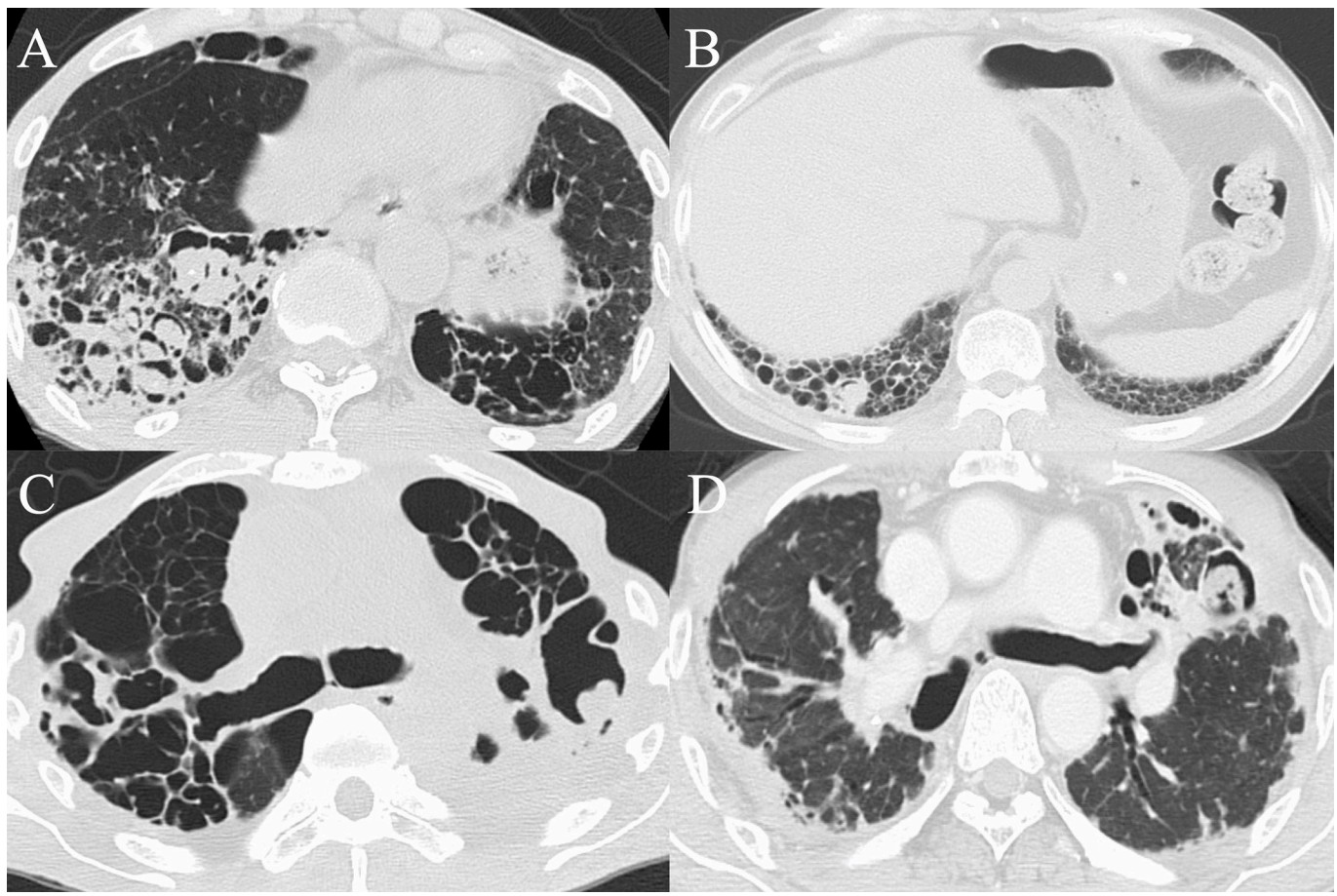

**Fig 3. Axial computed tomography images showing one or more aspergillomas within interstitial lung abnormalities findings.** A. multiple aspergillomas detected in the right lower lobe fibrosis in a combined pulmonary fibrosis and emphysema patient. B. One aspergilloma detected in the right lower lobe honeycombing (UIP). C. one aspergilloma in the left upper lobe fibroelastosis (PPFE pattern). D. A similar finding as Fig 3C in another patient.

0.005), presence of aspergilloma (1.59, 1.04–2.45, p = 0.033), and lower serum albumin (0.56, 0.38–0.83, p = 0.004), were independent predictors of mortality (Table 3).

## Antifungal treatments

Of the 264 patients, 190 (72.0%) received antifungal treatment regardless of the duration of treatment. Maintenance therapy was administered to 179 (94.2%) patients, and the breakdown of the first line maintenance therapy were as follows: itraconazole capsule 60 (33.5%), itraconazole oral solution 78 (43.6%), and voriconazole 39 (21.8%). In a comparison between antifungal-treated and untreated patients, age did not differ (treated: 70.3±10.9, untreated: 72.7±11.5, p = 0.11), but the NTM rates in untreated patients were statistically higher (treated: 26.1%, untreated: 50.0%, p<0.001). (See S1 Table) The 2-year survival rates in the treated and untreated patients was 67.9±3.7%, 74.6±5.8%, respectively and not different (p = 0.80). The reasons for why antifungal agents were not administered in 74 patients were as follows: priority of NTM treatment (38 patients), stable status (7 patients), priority of treatment for other diseases (5 patients), financial hardship (2 patients), and patient preference (2 patients), and unknown (20 patients).

**Table 3. Analysis of prognostic factors related to mortality in patients with chronic pulmonary aspergillosis.**

| | Univariate | | Final multivariate | |
|---|---|---|---|---|
| | Hazard ratio (95% CI) | p-value | Hazard ratio (95% CI) | p-value |
| Age | 1.05 (1.03–1.07) | <**0.001** | 1.05 (1.02–1.08) | <**0.001** |
| Male sex | 2.31 (1.37–3.89) | **0.002** | 2.48 (1.34–4.59) | **0.004** |
| Body mass index < 18.5 kg/m² | 1.88 (1.25–2.83) | **0.002** | 1.87 (1.20–2.90) | **0.005** |
| Underlying pulmonary conditions | | | | |
| Previous pulmonary tuberculosis | 0.91 (0.61–1.34) | 0.62 | 1.22 (0.76–1.95) | 0.42 |
| Non-tuberculous mycobacterium | 1.05 (0.70–1.58) | 0.81 | 0.85 (0.52–1.38) | 0.51 |
| Emphysema | 1.00 (0.67–1.47) | 0.98 | 0.69 (0.44–1.06) | 0.092 |
| Pneumothorax | 1.63 (0.66–4.00) | 0.29 | 0.96 (0.34–2.70) | 0.94 |
| Bronchiectasis | 0.71 (0.41–1.23) | 0.22 | 1.43 (0.74–2.75) | 0.28 |
| Thoracic surgery | 1.22 (0.68–2.18) | 0.51 | 1.04 (0.55–1.95) | 0.91 |
| Interstitial lung abnormalities | 2.36 (1.55–3.58) | <**0.001** | 1.76 (1.06–2.92) | **0.029** |
| Presence of aspergilloma | 1.55 (1.07–2.25) | **0.021** | 1.59 (1.04–2.45) | **0.033** |
| Smoking history | 1.62 (0.99–2.65) | 0.054 | - | |
| Auto Immune Diseases | 1.71 (0.79–3.71) | 0.17 | - | |
| Use of systemic corticosteroids | 2.84 (1.50–5.38) | **0.001** | 1.93 (0.91–4.09) | 0.085 |
| Albumin | 0.40 (0.30–0.53) | <**0.001** | 0.56 (0.38–0.83) | **0.004** |
| C-reactive protein | 1.06 (1.04–1.09) | <**0.001** | 1.02 (0.99–1.06) | 0.23 |

## Discussion

We presented the results from a prognostic analysis of 264 patients from an experienced single center in Japan regarding pulmonary aspergillosis. The 1-, 3-, and 5-year survival rates for the 264 patients were 78.7%, 61.0%, and 47.4%, respectively. Compared to the largest retrospective cohort of 387 CPA patients from the UK [8], which showed 1-, 3-, and 5-year survival rates of 86%, 75%, and 62%, respectively, the survival rates in this cohort were lower. But the second largest prognosis cohort of 194 CPA patients from Japan presented 1-, 3-, and 5-year survival rates of 80%, 67%, and 49% respectively [5], which were similar to our study. The mean age was 59.4 years in the UK study, while in the previous Japanese study and our study the ages were 68.5 and 71.0 years, which may account for the discrepancy in the results.

It is noteworthy that the most common underlying pulmonary condition was previous pulmonary tuberculosis (59.1%), and ILA were found in about 20% of CPA patients. There were regional differences of underlying pulmonary conditions in CPA patients: the most common condition was previous tuberculosis in Asian countries including Japan, but in contrast, that was chronic obstructive pulmonary disease in European countries. Curiously, sarcoidosis or allergic bronchopulmonary mycosis were seldom reported from Asia. The high prevalence rate of previous tuberculosis in our cohort reflects that Japan had a high incidence rate of tuberculosis after World War II, and is still a tuberculosis middle-burden country, with a notification rate of 13.3 cases per 100000 population in 2017 [23]. Even though previous reports showed 2–17% complicated-ILD in CPA patients [3, 5, 6, 11, 12], a few reports have pointed out that ILA/ILD can be a potential predisposition for the development of CPA, most of which were from Japan [5, 24]. In the present study, one or more aspergillomas, namely solid evidence of CPA, were found in cysts related to ILA in 60% of ILA-complicated cases. It is very interesting to consider whether this phenomenon is unique to Japan.

We found that underlying ILA was associated with poor prognosis, which remained significant even in multivariate analysis. In the previous cohort of 194 patients from Japan [5], 32 patients (16.5%) were complicated with ILD, and those with ILD had a significantly poorer

prognosis than with previous tuberculosis (the most common group). Although this result and ours seemed natural in consideration of the poor prognosis of patients with ILD, the high prevalence rate of ILA/ILD is notable, and it is still important to pay more attention to hidden aspergilloma in case of ILA/ILD because it is likely to be overlooked. The effect of complicating CPA on the prognosis of patients with ILD has rarely been explored, but there is a retrospective cohort of 539 patients with ILD which reported that 15 (2.9%) patients developed CPA in 44.0±37.8 months, and that in the CPA-complicated group, the rates of deaths, emphysema, and home oxygen therapy (HOT) usage were significantly higher [24]. Further studies about the complication of CPA and ILA/ILD, should examine whether complicating CPA has a negative impact on the prognosis of patients with ILA/ILD.

NTM-PD was detected in one third of the patients and not associated with a poorer prognosis for CPA patients even though about 40% of CPA patients with NTM-PD were not under CPA treatment. This corresponded to the prior report comparing the co-infection group (CPA with NTM) with the CPA without NTM group, which showed that there were no significant differences in cumulative survival rate between the groups (by log-rank test, p = 0.76) and that the rate of CPA treatment initiation in the co-infection group was significantly lower than in the CPA without NTM group (33.3% vs. 84.4%, p = 0.006) [15]. Co-infection of those infectious diseases have several problems. One noteworthy challenge is that treating both infections is often troublesome due to the interaction of antimicrobial agents. When the treatment of both infections is difficult for some reason, which treatment should be given priority may depend on the activity of NTM and CPA (possibly reflected as sputum culture status or anti-Aspergillus antibody titer [25, 26]), and also the etiologic organism of NTM. Even though the prognostic value of the etiologic NTM organism on NTM-complicated CPA patients has rarely been investigated, a recent cohort of 1,445 NTM patients from Korea reported that etiologic NTM organisms were significantly correlated with mortality: compared to *M. avium*, *M. intracellulare* (adjusted HR 1.40, 95%CI 1.03–1.91), *M. abscessus* (2.19, 1.36–3.51), and *M. massiliense* (0.99, 0.61–1.64) [13]. The etiologic NTM organism is also likely to be associated with mortality of NTM-complicated CPA patients, although this did not reach statistical significance in our analysis. Contrary to our results, in a previously noted cohort of 384 CPA patients from the UK [8], where about 10% of patients had NTM, underlying NTM-PD was an independent prognostic factor (adjusted HR2.07, 95%CI 1.22–3.52. Details of the etiologic organism of NTM infection were not described). In consideration of regional differences in NTM etiologic organisms [27] and possible effects of NTM etiologic organisms on CPA prognosis, the differences may be due to variations in etiologic NTM organisms. We frequently perform a sputum test for NTM-suspected patients and can easily perform an anti-Aspergillus antibody at our hospital, so it is also possible that we picked up more stable CPA patients with NTM-PD when considering the difference in the ratio of NTM-complicated patients. (33% vs. 10%) Complication of CPA and NTM should be surveyed further focusing on the etiologic NTM organisms and the activities of both infections.

In this analysis, multivariate Cox proportional hazards modeling showed that older age, male sex, BMI of <18.5 kg/m$^2$, serum albumin, and the presence of aspergilloma to be negative prognostic factors. Although not previously reported, male sex was an independent prognostic factor for CPA. We have no explanation for why males would die more frequently than females, although this association has also been noted for pulmonary tuberculosis [28] and also NTM [13, 29, 30]. Our results also confirmed previous findings that BMI of <18.5 kg/m$^2$ [6, 11], lower serum albumin [8], and the presence of aspergilloma [8], were independent risk factors for mortality.

There are some limitations to the present study. First, a certain number of patients did not receive antifungal treatments for various reasons and the choice of antifungal treatment and

its duration were dependent on attending physicians. There was however no difference in survival rates between treated and untreated patients, which implies that those untreated had mild disease or the priority was on other conditions like NTM. Second, we could not analyze causes of death due to incomplete data and difficulty in judging such causes. Especially, we were very interested in the causes of death in CPA patients complicated with ILA, but in many cases it was difficult to identify the causes because both CPA and ILA gradually got worse. Finally, we could assess only seven categories we adopted as underlying pulmonary conditions of CPA. But in consideration of the regional differences of the underlying pulmonary conditions, it is impossible to compare all the possibilities at the same time.

## Conclusion

Approximately half of our CPA patients had died in five years from the diagnosis. Several negative prognostic factors were clarified: interstitial lung abnormalities, older age, male sex, lower body mass index, the presence of aspergilloma, lower serum albumin. Focusing on such factors, early diagnosis and intervention is required.

## Supporting information

**S1 Fig. Patient flow.** We screened patients with positive serum anti-*Aspergillus* antibody obtained between January 2012 and April 2017. After applying the exclusion criteria, we enrolled 264 newly-diagnosed chronic pulmonary aspergillosis patients.
(TIF)

**S2 Fig. Survival of patients complicated with *M.avium* (n = 38), *M.intracellulare* (n = 20), *M.abscessus* complex (n = 5), and others (n = 24).**
(TIF)

**S1 Table. Background characteristics grouped by anti-fungal drug administration.**
(DOCX)

## Acknowledgments

The authors would like to thank Manabu Akazawa (Department of Public Health and Epidemiology, Meiji Pharmaceutical University) for support with statistical analysis. We also thank the laboratory staff at our hospital for the extracting necessary information for this paper.

## Author Contributions

**Conceptualization:** Yuya Kimura.

**Formal analysis:** Yuya Kimura, Yuka Sasaki.

**Investigation:** Yuya Kimura, Yuka Sasaki, Jun Suzuki, Hiroshi Igei.

**Methodology:** Yuya Kimura, Yuka Sasaki, Hirotoshi Matsui.

**Software:** Yuya Kimura.

**Supervision:** Junko Suzuki, Hirotoshi Matsui.

**Validation:** Yuya Kimura, Yuka Sasaki, Hirotoshi Matsui.

**Writing – original draft:** Yuya Kimura.

**Writing – review & editing:** Yuka Sasaki, Junko Suzuki, Maho Suzukawa, Hirotoshi Matsui.

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
