## [Decision Letter · Decision Letter 0]

10 Mar 2021

PONE-D-20-38958

Prognostic factors of chronic pulmonary aspergillosis: A retrospective cohort of 264 patients from Japan

PLOS ONE

Dear Dr. Kimura,

Thank you for submitting your manuscript to PLOS ONE. After careful consideration, we feel that it has merit but does not fully meet PLOS ONE’s publication criteria as it currently stands. Therefore, we invite you to submit a revised version of the manuscript that addresses the points raised during the review process.

We look forward to receiving your revised manuscript.

Kind regards,

Dr Aleksandra Barac

Academic Editor

PLOS ONE

Journal Requirements:

Reviewers' comments:

Reviewer's Responses to Questions

**Comments to the Author**

1. Is the manuscript technically sound, and do the data support the conclusions?

Reviewer #1: Yes

Reviewer #2: Yes

2. Has the statistical analysis been performed appropriately and rigorously? 

Reviewer #1: I Don't Know

Reviewer #2: Yes

3. Have the authors made all data underlying the findings in their manuscript fully available?

Reviewer #1: Yes

Reviewer #2: Yes

4. Is the manuscript presented in an intelligible fashion and written in standard English?

Reviewer #1: Yes

Reviewer #2: No

5. Review Comments to the Author

Reviewer #1: The authors perform a well-designed works. Only, I have some questions:

• Do you assessed for Pneumocystis jirovecii coinfection?

• What is your suggestion for prevention of CPA in patients with underling diseases?

• What were your lab test for diagnosis of active cases of CPA? Only antibody?

Reviewer #2: In their manuscript, the authors aim to identify factors that have an influence on mortality in a cohort of 264 CPA patients, focusing on underlying diseases as markers of prognosis.

The main finding of the work is the identification of interstitial lung disease / abnormalities (ILD/ILA) as a statistically significant prognostic marker. On the other hand, NTM-PD and other underlying diseases did not influence overall survival.

The cohort size is considerable; the results are interesting and deserve being reported.

However, a better definition of different CPA patterns and a more differentiated look at the underlying ILD are necessary.

Major points:

1. The different patterns of CPA in this cohort should be described.

Rationale: CPA is in itself heterogeneous, with its generally accepted patterns being Aspergilloma, Aspergillus nodule, CCPA, CFPA (not included here is SAIA, with overlapping features of IA).

These patterns probably describe a continuum of CPA disease, with CFPA, also sometimes referred to as „destroyed lung“, being the most severe, with extensive secondary fibrosis. Thus, when exploring prognostic factors in CPA, describing the CPA patterns is important.

2. The specific forms of the underlying ILD should be described.

Rationale: ILD is an umbrella term used to describe „diffuse parenchymal lung disease“, eventually leading to the development of pulmonary fibrosis. Etiology and prognosis are highly variable. Therefore, in the context of this paper, it is important to understand which forms of ILD were present in the CPA cohort. The type of ILD, such as Idiopathic pulmonary fibrosis (IPF), may determine an individual patient`s prognosis rather than the occurrence of CPA.

In this study, the radiological findings „honeycombing”, “upper lobe fibroelastosis” and “lower lobe fibrosis“, are used to define ILA. However, all of these changes also occur be secondary to CPA and therefore may present disease progression rather than ILD as an underlying disease.

The HRCT pattern of ILD should be described more specifically, (e.g. „UIP“, „NSIP“) and a clear definition of the underlying ILD should be attempted, such as:

• ILD with known etiology, such as ILD associated with systemic rheumatic disease

• ILD with unknown etiology such as Idiopathic interstitial pneumonia, e.g. IPF or Non-Specific Interstitial Pneumonia (NSIP)

• Others (e.g. ILD featuring cysts, such Pulmonary Langerhans cell histiocytosis or Pulmonary lymphangioleiomyomatosis)

• CPFE (Combined pulmonary fibrosis and emphysema)

Minor points:

In general, the discussion could be improved, incorporating the major points mentioned above. There are several minor points that should be addressed by the authors. Throughout the manuscript, the terms „complicating“ and „underlying“ seem to be used interchangeably, which is a bit misleading in some sentences. A professional language editing service may be considered to improve expression and clarity.

Abstract (page 2)

Line 27: Should read „we conducted a retrospective cohort study of…

Line 35: Should read „NTM-PD was not associated with higher mortality“, as this would be the logical hypothesis

Introduction:

Page 3:

Line 46: should read „Patients with CPA have a variety of underlying pulmonary conditions“, because CPA is usually the complication of other lung diseases and not vice versa

Line 50 ff.: This sentence merely repeats the statement of the previous sentences

Page 4:

Line 55: should read „were associated with ILD“

Line 56: should read „underlying NTM-PD“

Page 5:

Lines 75-77: should read „We had limited CPA cases to cases with a confirmed…“, otherwise the sentence gets the wrong meaning

Results:

Page 13

Lines 210, 215, Table 3: serum albumin and CrP: although it may be obvious, it should be mentioned that low albumin and elevated CrP (which cut-offs were used?) were prognostic markers, not just albumin and CrP

Discussion:

Page 16:

Line 264: should read „we found that underlying ILA was associated“

Page 17:

Line 274: the acronym „HOT” is used for the first time here and should be explained“

Line 278 ff.: As a possible explanation of these findings, it should be discussed that a subset of the CPA cases reported to complicate NTM-PD could be in fact aspergillus colonization.

Given the overlap in both symptoms and radiological appearance between NTM-PD and CPA, colonization and infection are particularly hard to differentiate.

6. PLOS authors have the option to publish the peer review history of their article (what does this mean?). If published, this will include your full peer review and any attached files.

Reviewer #1: **Yes: **Amir Abdoli

Reviewer #2: No

---

## [Author Response · Author response to Decision Letter 0]

17 Mar 2021

Dear Editor and Reviewers

 March 16th, 2021

Thank you very much for reviewing our manuscript and offering valuable advice.

We have addressed your comments with point-by-point responses, and revised the manuscript accordingly.

To facilitate your review of our revisions, the following is a point-by-point response to the questions and comments delivered in your letter dated March 11th, 2021.

REVIEWER #1 COMMENTS:

1. Do you assessed for Pneumocystis jirovecii coinfection?

RESPONSE: 

No patient in this study had image findings typical of Pneumocystis pneumonia (PCP), so we did not perform bronchoalveolar lavage test for PCP. In some cases, the elevation of serum Beta-D-glucan was detected, but we think it was attributable to CPA. 

2. What is your suggestion for prevention of CPA in patients with underling diseases?

RESPONSE:

Since Aspergillus species are common in our surrounding environment, it is difficult to completely prevent contacting with them. But it may be useful to avoid circumstances filled with mold or to wear a mask in such circumstances in order not to develop CPA.

3. What were your lab test for diagnosis of active cases of CPA? Only antibody?

RESPONSE:

We usually perform the following tests for assessing CPA: serum Beta-D-glucan and anti-Aspergillus antibody, bacteriological inspections (of sputum/ samples from bronchoscopy), and pathological examination (of sputum/ samples from bronchoscopy). We have limited CPA patients to those with confirmed positive serum anti-Aspergillus antibody to secure homogeneity.

REVIEWER #2 COMMENTS:

1. The different patterns of CPA in this cohort should be described.

Rationale: CPA is in itself heterogeneous, with its generally accepted patterns being Aspergilloma, Aspergillus nodule, CCPA, CFPA (not included here is SAIA, with overlapping features of IA).

These patterns probably describe a continuum of CPA disease, with CFPA, also sometimes referred to as „destroyed lung“, being the most severe, with extensive secondary fibrosis. Thus, when exploring prognostic factors in CPA, describing the CPA patterns is important.

RESPONSE:

As you mentioned, there are many patterns of Aspergillus infectious diseases: aspergilloma, Aspergillus nodule, SAIA, CCPA, and CFPA. In this study, aspergilloma and Aspergillus nodule cases were not included because of the inclusion criteria (We added some explanations for clarity in the manuscript). And we did not distinguish between CCPA and CFPA because it is actually difficult to do it, and because the distinction has not been made in most of recent CPA prognostic studies.

2. The specific forms of the underlying ILD should be described.

Rationale: ILD is an umbrella term used to describe „diffuse parenchymal lung disease“, eventually leading to the development of pulmonary fibrosis. Etiology and prognosis are highly variable. Therefore, in the context of this paper, it is important to understand which forms of ILD were present in the CPA cohort. The type of ILD, such as Idiopathic pulmonary fibrosis (IPF), may determine an individual patient`s prognosis rather than the occurrence of CPA.

In this study, the radiological findings „honeycombing”, “upper lobe fibroelastosis” and “lower lobe fibrosis“, are used to define ILA. However, all of these changes also occur be secondary to CPA and therefore may present disease progression rather than ILD as an underlying disease.

The HRCT pattern of ILD should be described more specifically, (e.g. „UIP“, „NSIP“) and a clear definition of the underlying ILD should be attempted, such as:

• ILD with known etiology, such as ILD associated with systemic rheumatic disease

• ILD with unknown etiology such as Idiopathic interstitial pneumonia, e.g. IPF or Non-Specific Interstitial Pneumonia (NSIP)

• Others (e.g. ILD featuring cysts, such Pulmonary Langerhans cell histiocytosis or Pulmonary lymphangioleiomyomatosis)

• CPFE (Combined pulmonary fibrosis and emphysema)

RESPONSE:

Thank you for providing these insights. In consideration that few cases had undergone serum tests for assessing the type of ILA/ILD and pathological examination (of specimens from bronchoscopy/lung surgery), we believe a clear definition of the underlying ILD is difficult. 

Following your advice, we have assessed the HRCT patterns of ILA/ILD more precisely by making five categories: ①UIP/②NSIP/③PPFE pattern/④combined pulmonary fibrosis and emphysema/⑤unclassifiable. The breakdown of the categories was added into the RESULTS section. We have not performed survival analysis grouped by the categories because the sample size of each category is too small.

3. Minor points

RESPONSE:

We really appreciate your pointing out our unsuitable expressions in detail. We have dealt with them one by one. Some points we would like to give comments were as below.

Line 50 ff.: This sentence merely repeats the statement of the previous sentences

・RESPONSE: In this sentence, we want to clarify that most common underlying pulmonary conditions in CPA patients are COPD and tuberculosis.

Results:

Page 13

Lines 210, 215, Table 3: serum albumin and CrP: although it may be obvious, it should be mentioned that low albumin and elevated CrP (which cut-offs were used?) were prognostic markers, not just albumin and CrP

・RESPONSE: We have corrected “serum albumin, C-reactive protein” to “lower serum albumin, elevated C-reactive protein”. The linearity of the relationships between the outcome and serum albumin/C-reactive protein was not problematic, so we did not adopt cut-offs for them.

Again, we appreciate all of your insightful comments. We worked hard to be responsive to them. Thank you for taking the time and energy to help us improve the paper.

Sincerely,

Yuya Kimura, M.D.

National Hospital Organization Tokyo National Hospital,

3-1-1 Takeoka, Kiyose-shi, Tokyo, 204-8585, Japan

Tel. +81-42-491-2111, Fax. +81-42-494-2168

E-mail. yuk.close.to.wrd.34@gmail.com

---

## [Editor Report · Decision Letter 1]

19 Mar 2021

Prognostic factors of chronic pulmonary aspergillosis: A retrospective cohort of 264 patients from Japan

PONE-D-20-38958R1

Dear Dr. Kimura,

We’re pleased to inform you that your manuscript has been judged scientifically suitable for publication and will be formally accepted for publication once it meets all outstanding technical requirements.

Kind regards,

Dr Aleksandra Barac

Academic Editor

PLOS ONE

---

## [Editor Report · Acceptance letter]

23 Mar 2021

PONE-D-20-38958R1 

Prognostic factors of chronic pulmonary aspergillosis: A retrospective cohort of 264 patients from Japan 

Dear Dr. Kimura:

I'm pleased to inform you that your manuscript has been deemed suitable for publication in PLOS ONE. Congratulations! Your manuscript is now with our production department. 

Kind regards, 

on behalf of

Dr. Aleksandra Barac 

Academic Editor

PLOS ONE